# Metabolomics for prediction of hypertension in pregnancy: a systematic review and meta-analysis protocol

Jussara Mayrink,[1] Debora Farias Batista Leite ![ORCID] ,[1,2] Maria Laura Costa,[1] Jose Guilherme Cecatti ![ORCID] [1]

[1]Department of Gynecology and Obstetrics, State University of Campinas, Campinas, Brazil
[2]Department of Maternal and Child Health, Federal University of Pernambuco, Recife, Brazil

**Correspondence to**
Professor Jose Guilherme Cecatti; cecatti@unicamp.br

## ABSTRACT

**Introduction** Hypertension is a very important cause of maternal morbidity and mortality worldwide, despite efforts on prevention. The lack of a tool to provide effective and early prediction of hypertension for a high-risk group may contribute to improving maternal and fetal outcomes. Metabolomics has figured out as a promised technology to contribute to the improvement of hypertension in pregnancy prediction.

**Methods and analysis** Our primary outcome is hypertensive disorders of pregnancy. A detailed systematic literature search will be performed in electronic databases PubMed, EMBASE, Scopus, Web of Science, Latin America and Caribbean Health Sciences Literature, Scientific Electronic Library Online, Health Technology Assessment and Database of Abstracts of Reviews of Effects using controlled terms 'pre-eclampsia', 'hypertensive disorders', 'metabolomics' and 'prediction' (and their variations). Studies from the latest 20 years will be included, except case reports, reviews, cross-sectional studies, letter to editors, expert opinions, commentaries papers or non-human research. If possible, we will perform a meta-analysis. Two peer-reviewers will independently perform the search and in cases of discordance, a third reviewer will be consulted.

**Ethics and dissemination** As a systematic review, ethics approval is not required. The results of this review will present the current use and performance of metabolomics for predicting gestational hypertension. Such data could potentially guide future studies and interventions to improve existing prediction models.

**PROSPERO registration number** CRD42018097409.

## INTRODUCTION

Hypertensive disorders in pregnancy consist of a group of conditions including pre-eclampsia, gestational hypertension, pre-eclampsia superimposed to chronic hypertension, white coat hypertension, masked hypertension and transient hypertension[1,2] and appear as the second cause of maternal death in the world according to a study performed by WHO between 2003 and 2009.[3] Pre-eclampsia is the leading cause of maternal morbidity and mortality in Brazil and several other low-income and middle-income

---

**STRENGTHS AND LIMITATIONS OF THIS STUDY**

⇒ Electronic search will cover the most important current available scientific databases for health research.
⇒ There will not be a language restriction.
⇒ Considering the complexity of metabolomics technology and its methods, there would be a limitation to perform a quantitative synthesis.

---

countries.[4,5] Its prevalence can vary according to the set of analyses, but the number ranges from 2% to 10% of all pregnancies.[4] Every year, around 70 000 women die because of pre-eclampsia and its complications,[3] despite potential prevention implemented by low-dose aspirin.[6,7] This intervention can represent a reduction rate of around 50% in the incidence of the early-onset pre-eclampsia cases, which developed pre-eclampsia before 34 weeks of gestation.[7,8] In this scenario, the prediction of pregnant women under high risk to develop pre-eclampsia is a key topic.

Some biomarkers have been proposed as earlier predictors (placental growth factor, pregnancy-associated plasma protein A) combined with clinical factors (pulsatility index of uterine arteries at Doppler velocimetry examination, mean arterial blood pressure) in models with different detection and false-positive rates.[9–12] These studies present limitations regarding the number of participants enrolled and heterogeneity to assess the prediction performance of those factors. Furthermore, the proposed prediction models from combining those factors outline better detection rates for early-onset pre-eclampsia cases compared with late-onset cases.[13–15]

In the last decade, with the broad application of omics technologies, metabolomics has been pointed as a promising tool for the identification of early predictors for many health disturbances[16–18] and pre-eclampsia is one of

them. Through metabolomics, it would be possible to identify metabolites involved in the final line of gene expression and a phenotypic signature in high resolution of the disease to be studied.[19–21] Studies have provided some insights about pre-eclampsia prediction through metabolites, belonging to different chemical classes and showing different performances.[20–23] Kenny *et al* provided the initial knowledge on the topic, identifying 14 metabolites belonging to different chemical classes. When combined in an algorithm, they showed a very good performance, with an area under the curve of 0.94 in a discovery phase of the study and a detection rate of 77%, considering a false-positive rate of 10%.[22] It represents a very important tool option for prediction, especially concerning cases of late-onset pre-eclampsia, which are the majority and the most difficult cases to predict.[13–15] Thus, in the sense of the inexistence of a systematic review protocol registered in this topic as well as a systematic review in progress or published, the main objective of this systematic review is to determine the accuracy of metabolomics for predicting hypertensive disorders of pregnancy.

## QUESTION FORMULATION

Because of the social and economic implication of hypertensive disorders, their consequences to maternal and fetal lives worldwide and the lack of a useful screening test, in parallel to the increase of applicability of omics technologies, this systematic review will be guided by this question: what is the performance of metabolomics for predicting gestational hypertensive disorders? It is following the PICO method[24] and associated with the search strategy provided a preliminary flow chart of studies as summarised in figure 1.

## METHODS AND ANALYSIS
### Search strategy

Electronic searches of literature will be carried out with these following databases: PubMed, EMBASE, Scopus, Web of Science, Latin America and Caribbean Health Sciences Literature, Scientific Electronic Library Online (Scielo), Health Technology Assessment, Database of Abstracts of Reviews of Effects. We will include studies from the latest 20 years, considering that the vast majority of manuscripts on metabolomics are from this century. Our search strategy will combine terms with Boolean connectors related to the following categories: (1) hypertensive disorders, pre-eclampsia, pregnancy; (2) metabolomics, metabolome and (3) screening, prediction. The Boolean connectors will be adapted according to the database used. We decided to use regular terms—not MeSH or Emtree terms—taking into account the number of databases consulted, to use always the same terms for all of them. Also, we will search reference list of included articles, doing the backtracking of references. There will not be a language restriction. Before final publication, we will perform a new search in the databases to check if any

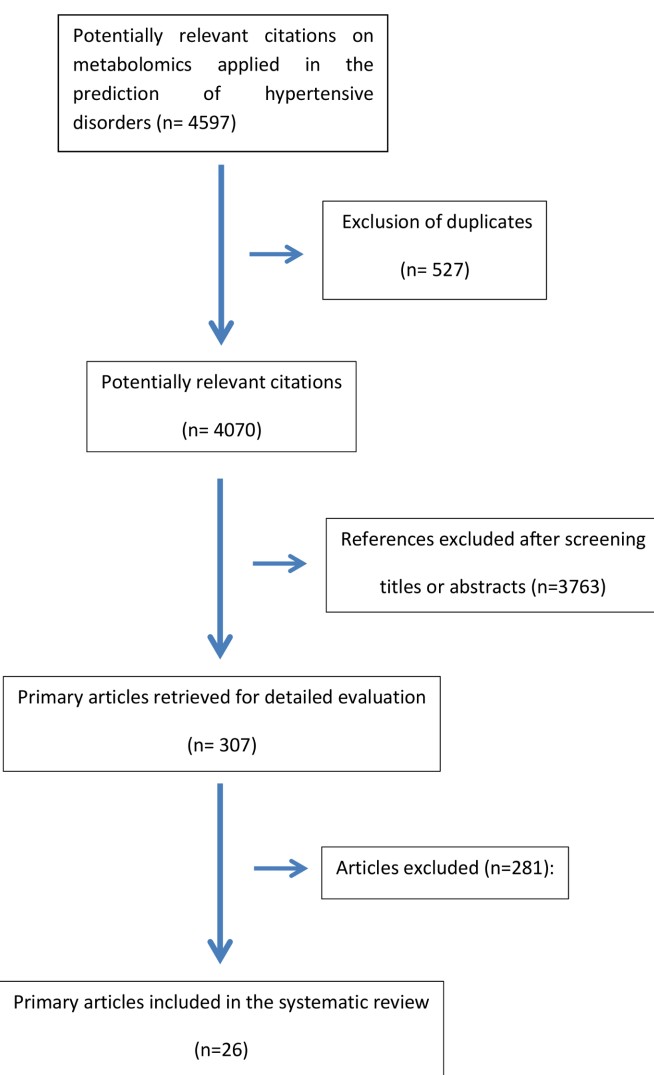

**Figure 1** Flow chart of studies identified to be included in the systematic review.

study was published during the period of the systematic review elaboration. The databases exploration process and its results will follow the Preferred Reporting Items for Systematic Reviews and Meta-Analyses (PRISMA) statement.[25]

### Study selection process

After searching all sources of databases cited above, all the citations will be exported into EndNote software. First, two reviewers (JM and DFBL) will independently assess titles and abstracts. Only papers considered potentially relevant according to the inclusion criteria will be retrieved for further consideration. Cases of divergence will be analysed by a third reviewer (MLC) who will do the final decision. A fourth reviewer (JGC) will check all procedures before approving the data extraction.

### Study inclusion criteria

Hypertensive disorders developed at any gestational age will be considered the domain studied. Previous other chronic conditions (diabetes, renal diseases, etc) will be

reported for stratification of analysis if the data allow for this. Original studies—including diagnostic studies—involving pregnant women are the inclusion criteria, and congenital malformation is the exclusion criteria.

## Interventions/exposure

Prediction of hypertensive disorders through metabolomics technologies is the intervention to be studied. The biomarker analysis should have been performed on samples taken before the hypertensive disorder diagnosis.

## Design

Our systematic review will include original studies (cohort or case–control studies), including single or multiple pregnancies, as the studied population, and hypertensive disorders developed at any time of pregnancy, as the outcome of interest. We will exclude any studies that are: cross-sectional studies, case reports, editorials, letter to editors, commentaries, expert opinions, any type of reviews, and experimental studies with animals, and when it is not possible to extract the data about the outcomes of interest.

## Outcomes

We will include studies reporting outcomes of any hypertensive disorder developed during the pregnancy. Our primary outcome is pre-eclampsia, defined as the onset of hypertension (systolic blood pressure of 140 mm Hg or more and/or diastolic blood pressure of 90 mm Hg or more) after 20 weeks of gestation, measured at least in two different occasions, combined with (1) proteinuria (300 mg/day or at least 1 g/L (1+) on dipstick testing or spot urine protein/creatinine >30 mg/mmol (0.3 mg/mg)) or (2) systemic complications or (3) uteroplacental dysfunction (fetal growth restriction).[1] By systemic complications, we will consider:
► Haematological complications (thrombocytopaenia—platelet count below 150 000/dL, disseminated intravascular coagulation, haemolysis).
► Hepatic dysfunction (elevated transaminases—at least twice upper limit of normal +− right upper quadrant or epigastric abdominal pain).
► Neurological dysfunction (examples include eclampsia, altered mental status, blindness, stroke or more commonly hyper-reflexia when accompanied by clonus, severe headaches when accompanied by hyper-reflexia, persistent visual scotomata).
► Renal dysfunction (creatinine >1.2 mg/dL).
  Secondary outcomes include:
► Early-onset pre-eclampsia: when occurs before or at 33 weeks of gestation.[26]
► Late-onset pre-eclampsia: when occurs at or after 34 weeks of gestation.[26]
► Gestational hypertension: de novo development of high blood pressure after 20 weeks of gestation (systolic blood pressure of 140 mm Hg or more and/or diastolic blood pressure of 90 mm Hg or more),

without any of the abnormalities that define pre-eclampsia as discussed above.[1]
► Whitecoat hypertension: it is demonstrated when normal blood pressure is registered during 24 hours ambulatory monitoring in the first half of pregnancy.[1]
► Pre-eclampsia superimposed on chronic hypertension: in a patient with high blood pressure predating the pregnancy, it is registered the occurrence of pre-eclampsia.[1]
► Masked hypertension: is characterised by blood pressure that is normal at office or clinic but elevated at other times, most typically diagnosed by 24 hours ambulatory blood pressure monitoring.[2]
► Transient gestational hypertension is hypertension that arises in the second or third trimester. The hypertension is detected in the clinic but then settles with repeated blood pressure readings.[2]

## Data extraction

Data will be extracted through a standardised data compilation form in duplicate to avoid errors. The variables of interest from each included study are: authors, country, year of publication, study design, number of participants, pre-eclampsia prevalence, gestational age of recruitment, biological samples used, laboratory methods, metabolomics technology applied and metabolites. The metabolites will be matched with the Human Metabolome Database (HMDB) to check their biological function and chemical subclass. Missing data will be requested from study authors. Pairs of data-extraction forms will be checked for discrepancies.

## Quality appraisal

The same two reviewers (JM and DFBL) who judged eligibility of papers will independently assess the risk of bias in included studies, but this time rating the methodological quality of the primary research. A third reviewer (MLC) will solve divergences when needed. Quality Assessment of Diagnostic Accuracy Studies is the standard scale to be applied to access internal validity.[27] This tool is composed of four domains: patient selection, index test (metabolomics technique), reference standard (arterial blood pressure) and flow and timing of patient inclusion and follow-up. Each domain is assessed in terms of risk of bias and the first three are assessed in terms of concerns regarding applicability. For each domain, every study will be labelled as 'low', 'high' or 'unclear' risk of bias.

Funnel plots and sensitivity and cumulative analyses will be applied for the detection of temporal trends and publication bias.

## Strategy for data synthesis

Following the PRISMA, a flow diagram will be drawn.[25] Tables will show data regarding studies characteristics and risk of bias assessment for included and excluded studies. Narrative data will be analysed and structured according to the outcomes: pre-eclampsia, gestational hypertension, transient gestational hypertension, white

coat hypertension, masked hypertension. If possible, we are going to perform subgroup analysis according to the metabolomics methods applied: gas or liquid chromatography, coupled with mass spectrometry, or proton nuclear magnetic resonance, and based on ethnic group and the severity of the hypertensive disease. We also intend to perform a sensitivity analysis based on early and late pre-eclampsia cases if sufficient studies will be found.

A meta-analysis will be performed (hierarchical summary receiver characteristic operating curve) and accuracy measures will be calculated depending on data availability. If a meta-analysis will be possible, considering the limitations imposed by data heterogeneity and drawings of the vast majority of studies, we intend to use RevMan software. Taking into account that the studies involve the frequency of metabolites and occurrence of pre-eclampsia, we are going to use a fixed-effect model or random-effect model, depending on the heterogeneity found. Heterogeneity will also be assessed, through the $I^2$ test, Hotelling's $T^2$ test and Cochran's Q test.

## Ethics and dissemination

Prediction of hypertensive disorders has been studied over the years with specific challenges. Among nulliparous, for example, there is no history of previous events and a previous history of pre-eclampsia, is considered the most consistent predictive risk factor.[28] Another challenge to overcome is regarding late-onset pre-eclampsia cases, which represent the majority of them. As cited above, the algorithms composed by biochemical and clinical factors showed better results with early-onset cases of pre-eclampsia.[13 14]

Metabolomics is a very complex technology and it has emerged as a possibility for prediction of adverse pregnancy outcomes.[29–31] The techniques employed are nuclear magnetic resonance spectroscopy, gas or liquid chromatography-mass spectrometry, Fourier transforms infrared spectrometry and capillary electrophoresis.[31] Because of this complexity, results may be different concerning the metabolites found. Consequently, generalising results is also a challenge to overcome. This systematic review will contribute to optimise the knowledge about the metabolites found in the studies and perhaps classify them according to HMDB, enabling quality translational research.

Besides, this systematic review will contribute to establishing the current state of knowledge concerning the capacity of metabolomics to predict the occurrence of pre-eclampsia. Taking into account that this outcome involves relevant consequences for maternal and neonatal lives, the development of a tool that would predict pre-eclampsia is essential. Furthermore, the results of this systematic review could be used to guide future studies in this field. Once published, this systematic review will be freely available in an open-access scientific journal.

## Patient and public involvement

Patients or the public were not involved in the design, or conduct, or reporting, or dissemination plans of our research proposal.

**Acknowledgements** This is a modified version of the article that was part of the PhD thesis of Jussara de Souza Mayrink Novais presented to the Postgraduate Programme on Obstetrics and Gynecology from the School of Medical Sciences of the University of Campinas, Brazil, under the tutorial of Jose Guilherme Cecatti and Maria Laura Costa on 13 December 2018.

**Contributors** JM worked out the protocol, developed searches and data management, will participate in the selection, inclusion, quality assessment and data extraction of papers. DFBL helped working out the protocol and will participate in selection, inclusion, quality assessment and data extraction as well. MLC and JGC helped working out the protocol, and MLC will solve any disagreement concerning the selected papers. JGC will supervise all the development of the systematic review. All authors read and approved this final manuscript.

**Funding** This study is a subproduct of the study 'Preterm SAMBA' which was jointly financed by the Brazilian CNPq (CNPq, Grant 401636/2013-5) and the Bill & Melinda Gates Foundation (Grant OPP1107597).

**Competing interests** None declared.

**Patient consent for publication** Not required.

**Provenance and peer review** Not commissioned; externally peer reviewed.

**ORCID iDs**
Debora Farias Batista Leite http://orcid.org/0000-0001-8839-3934
Jose Guilherme Cecatti http://orcid.org/0000-0003-1285-8445

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
