## [Reviewer comments · BMJ Open]

ARTICLE DETAILS

TITLE (PROVISIONAL)	Metabolomics for prediction of hypertension in pregnancy: a systematic review and meta-analysis protocol
AUTHORS	Mayrink, Jussara; Leite, Debora Farias Batista; Costa, Maria Laura; Cecatti, Jose Guilherme

VERSION 1 – REVIEW

REVIEWER	Giuseppe Rizzo Università Roma Tor Vergata Italy
REVIEW RETURNED	24-Jun-2020

GENERAL COMMENTS	in this study Authors performed a systematic review and meta-analysis of the role of metabolomics in the prediction of hypertensive diseases. The subject is of interest and the study promises to be an important topic My only criticism that should be addressed is the difficulties in meta-analysis to properly classify the severity of the hypertensive diseases
---

REVIEWER	Anthony Au Universiti Sains Malaysia
REVIEW RETURNED	10-Oct-2020

GENERAL COMMENTS	This protocol paper will be focused on investigating the association between metabolomic biomarkers and gestational hypertension by using a meta-analysis technique. I have no major comments regarding the technical content, only few suggestions: 1. A more appropriate citation is required for “Every year, around 80 thousand 60 women die because of preeclampsia and its complications (6)”, as the cited reference did not contain the relevant information.2. The sentence “Some biomarkers have been proposed as earlier predictors (placental growth factor PIGF, pregnancy-associated plasma protein A-PAPP-A) combined with clinical factors (pulsatility index of uterine arteries at Dopplervelocimetry exam, mean arterial blood pressure), showing different and sometimes conflicting detection rates 9-12.” is not clear, and thus it should be amended.3. There are plenty of databases for meta-analysis data extraction, such as CNKI, IMSEAR, Jstage, KoreaMed, wangfang, and others, that are worth to be included.
---

	4. In addition to the subgroup analysis according to the metabolomics methods (gas or liquid chromatography, coupled with mass spectrometry, or proton nuclear magnetic resonance), subgroup analyses should be performed based on gender and ethnic groups. 5. Which software will be used for meta-analysis? STATA, RevMan, CMA or others? And which statistical models will be applied? 6. There are many grammatical errors throughout the manuscript and part of the language needs to be improved, in order to make better sense of what exactly you are trying to explain. I would recommend the manuscript to be revised by a native English speaker or alternatively by a commercial language editing service.
--	--

VERSION 1 – AUTHOR RESPONSE

Reviewer: 1

Comments to the Author

in this study Authors performed a systematic review and meta-analysis of the role of metabolomics in the prediction of hypertensive diseases. The subject is of interest and the study promises to be an important topic.

My only criticism that should be addressed is the difficulties in meta-analysis to properly classify the severity of the hypertensive diseases.

We appreciated your comment. Certainly, if a meta-analysis will be performed with information retrieved from studies, we intend to address this issue in sub-group analysis. This is now clearly stated in the Strategy for data synthesis item (lines 208-209). Also, otherwise we will consider all this limitation in the specific section.

Reviewer: 2

Comments to the Author

This protocol paper will be focused on investigating the association between metabolomic biomarkers and gestational hypertension by using a meta-analysis technique. I have no major comments regarding the technical content, only few suggestions:

1. A more appropriate citation is required for “Every year, around 80 thousand 60 women die because of preeclampsia and its complications (6)”, as the cited reference did not contain the relevant information.

Thank you for the comment. The adequacy of the reference list was done (lines 62-63).

2. The sentence “Some biomarkers have been proposed as earlier predictors (placental growth factor PIGF, pregnancy-associated plasma protein A-PAPP-A) combined with clinical factors (pulsatility index

of uterine arteries at Dopplervelocimetry exam, mean arterial blood pressure), showing different and sometimes conflicting detection rates 9-12.” is not clear, and thus it should be amended.

We totally agree with you, the sentence is not clear enough. The idea of this paragraph is to highlight to the readers that the studies involving biomarkers, mean arterial blood pressure, pulsatility index of uterine arteries at Dopplervelocimetry exam and their models (throughoutly explored) showed conflicting results concerning detection rate and false positive rate. The main reason to this is the heterogeneity of the sample size, as well as the number of enrolled participants. The corrections were done in the text (line 71).

3. There are plenty of databases for meta-analysis data extraction, such as CNKI, IMSEAR, Jstage, KoreaMed, wangfang, and others, that are worth to be included.

Thank you for your suggestion. According to what has already been accepted by the registration in the PROSPERO, the datasets to be accessed to get information for this systematic review are those stated in the search strategy session of the protocol (lines 105-108). In fact, there are several databases, therefore we choose those more frequently accessed to provide the required information in other systematic reviews.

4. In addition to the subgroup analysis according to the metabolomics methods (gas or liquid chromatography, coupled with mass spectrometry, or proton nuclear magnetic resonance), subgroup analyses should be performed based on gender and ethnic groups.

Thank you for this suggestion. Gender will be only female, considering we are focusing on pregnancy. The inclusion of ethnic group for subgroup analysis was done, besides the severity of hypertensive disease (lines 208-209).

5. Which software will be used for meta-analysis? STATA, RevMan, CMA or others? And which statistical models will be applied?

If a metanalysis will be possible, considering the limitations imposed by data heterogeneity and drawings of the vast majority of studies, we intend to use RevMan. Taking into account that the studies involve frequency of metabolites and occurrence of preeclampsia, we are going to use a fixed-effect model or random- effect model, depending on the heterogeneity found. Lines 212-217.

6. There are many grammatical errors throughout the manuscript and part of the language needs to be improved, in order to make better sense of what exactly you are trying to explain. I would recommend the manuscript to be revised by a native English speaker or alternatively by a commercial language editing service.

Thank you. We performed a complete review by a skilled professional on the language editing.